# Innovative Techniques for Infection Control and Surveillance in Hospital Settings and Long-Term Care Facilities: A Scoping Review

**DOI:** 10.3390/antibiotics13010077

**Published:** 2024-01-13

**Authors:** Guglielmo Arzilli, Erica De Vita, Milena Pasquale, Luca Marcello Carloni, Marzia Pellegrini, Martina Di Giacomo, Enrica Esposito, Andrea Davide Porretta, Caterina Rizzo

**Affiliations:** 1Department of Translational Research and New Technologies in Medicine and Surgery, University of Pisa, 56126 Pisa, Italy; guglielmo.arzilli@phd.unipi.it (G.A.); m.pasquale@studenti.unipi.it (M.P.); l.carloni1@studenti.unipi.it (L.M.C.); m.pellegrini1@studenti.unipi.it (M.P.); 23663031@studenti.unipi.it (M.D.G.); 23706116@studenti.unipi.it (E.E.); andrea.porretta@unipi.it (A.D.P.); caterina.rizzo@unipi.it (C.R.); 2University Hospital of Pisa, 56124, Pisa, Italy

**Keywords:** healthcare-associated infections, artificial intelligence, hospital, machine learning, surveillance, infection control

## Abstract

Healthcare-associated infections (HAIs) pose significant challenges in healthcare systems, with preventable surveillance playing a crucial role. Traditional surveillance, although effective, is resource-intensive. The development of new technologies, such as artificial intelligence (AI), can support traditional surveillance in analysing an increasing amount of health data or meeting patient needs. We conducted a scoping review, following the PRISMA-ScR guideline, searching for studies of new digital technologies applied to the surveillance, control, and prevention of HAIs in hospitals and LTCFs published from 2018 to 4 November 2023. The literature search yielded 1292 articles. After title/abstract screening and full-text screening, 43 articles were included. The mean study duration was 43.7 months. Surgical site infections (SSIs) were the most-investigated HAI and machine learning was the most-applied technology. Three main themes emerged from the thematic analysis: patient empowerment, workload reduction and cost reduction, and improved sensitivity and personalization. Comparative analysis between new technologies and traditional methods showed different population types, with machine learning methods examining larger populations for AI algorithm training. While digital tools show promise in HAI surveillance, especially for SSIs, challenges persist in resource distribution and interdisciplinary integration in healthcare settings, highlighting the need for ongoing development and implementation strategies.

## 1. Introduction

Addressing the challenge of healthcare-associated infections (HAIs) in the evolving healthcare landscape is paramount. HAIs significantly impact patient morbidity, mortality, and healthcare costs globally. According to the European Centre for Disease Prevention and Control (ECDC), the main types of HAIs are surgical site infections (SSIs), catheter-associated urinary tract infections (CAUTIs), central line-associated bloodstream infections (CLABSIs), ventilator-associated pneumonia (VAPs), and gastrointestinal infections. Among those, 16–20% include an emerging challenge due to the widespread and indiscriminate use of antibiotics for prophylactic or therapeutic purposes: multidrug-resistant organisms (MDROs) such as methicillin-resistant *Staphylococcus aureus* (MRSA), vancomycin-resistant *E. faecium* (VRE), carbapenem-resistant *P. aeruginosa*, and extended-spectrum cephalosporin-resistant *K. pneumoniae*. As reported by the WHO, out of every 100 patients in acute-care hospitals, 7 patients in high-income countries (HICs) and 15 patients in low- and middle-income countries (LMICs) will acquire at least one HAI during their hospital stay and, of those affected, 10% will die [1].

Most HAIs are preventable, and surveillance plays a role in reducing infections and controlling the emergence of multidrug-resistant microorganisms. It facilitates the identification of infection trends, informs prevention strategies, and assesses the effectiveness of interventions. However, traditional surveillance, mainly in its active form, is more time-consuming and costly in terms of resources, leading to potential delays and inaccuracies [2]. In fact, traditional surveillance requires dedicated teams of professionals to manually review patient’s medical records, applying standardized case definitions [3]. Also, outbreak detection is a critical component of infection control, requiring rapid identification and response. Conventional detection methods, relying on a healthcare personnel’s recognition of infection clusters, can be untimely and, especially in workforce shortage, unresponsive. Due to limited resources and poor availability of labour-intensive manual surveillance systems, most healthcare centres choose to apply “targeted forms” of surveillance, including, for example, only high-risk wards and/or a few types of medical procedures and/or only a few types of HAIs. The prevalence of HAIs is under-reported, mainly in developing countries, due to a lack of surveillance data [4].

The introduction of alternative methods using automated detection strategies, drawn on artificial intelligence (AI) and machine learning (ML) algorithms, seems promising to fulfil this gap and are now becoming increasingly popular in healthcare [5]. Automated surveillance can support (semi-automated) or completely replace (fully automated) manual surveillance thanks to the use of algorithms based on AI [6]. The development of new methods using artificial intelligence (AI) and machine learning (ML) algorithms can support or replace traditional surveillance because they can analyse a growing amount of health data made available through automated collection systems, an analysis that would otherwise require unsustainable amounts of time and human resources [7].

This scoping review first aims to delve into the latest advancements in infection control and surveillance within hospital settings and long-term care facilities, particularly emphasizing the critical role of innovative technologies in managing these infections and the need for robust protocols and research for their effective implementation. The second aim involves highlighting the strengths and weaknesses of the innovative techniques compared to traditional ones.

## 2. Methods

This scoping review has been conducted to explore innovative tools for HAI control and surveillance and their applications in hospital settings and long-term care facilities (LTCFs). The scoping review was conducted following the PRISMA-ScR guideline that ensures the transparency and completeness of the review process [8].

### 2.1. Search Strategy

We searched studies referring to new digital technologies applied to the surveillance, control, and prevention of HAIs in hospitals and LTCFs and published from 1 January 2018 to 4 November 2023. The search strategy was built on previously published reviews [9] and was conducted on PubMed, Scopus, and Web of Science for records reporting the following terms: (1) healthcare-associated infections, (2) hospital and LTCF settings, (3) surveillance, and (4) innovation technologies. We also checked reference lists of relevant systematic/scoping reviews for eligible studies. The specifics about search strategy are reported in the Appendix A.

### 2.2. Eligibility Criteria

For this review, we considered articles referring to innovative surveillance techniques, defining the latter as any new technology applied to the detection, control, and surveillance of HAIs in a real setting (we consider only practical application, not modelling) based on the categories reported by the ECDC Scoping Review [9]. Laboratory techniques and innovations related to clinical case management were excluded. For the secondary outcome of the review, we considered data on the evaluation and comparison of accuracy between innovative and traditional techniques for prevention, surveillance, and control of HAIs. We included randomized controlled trials, nonrandomized comparative studies, observational studies, and cross-sectional studies for the analysis. Reports of narrative reviews, case reports, and other non-pertinent publication types were excluded. We considered studies conducted worldwide, considering hospitalized patients and individuals admitted to long-term care facilities. No language restrictions were applied.

### 2.3. Study Selection

Eligible citations were downloaded and uploaded to bibliographic management web-application (Rayyan) and duplicates were removed. The selection procedure consisted of three phases: (1) a title abstract selection performed by seven reviewers, (2) a full-text screening of the selected articles performed by five reviewers, and (3) a final screening during the data-extraction phase performed to exclude articles that present identical outcome measures or have data not extractable. Study selection was based on the inclusion/exclusion criteria presented in Table 1.

Records were selected through an iterative double screening of a subset of records to achieve a concordance >95%, followed by a single screening of the remaining records. The group of reviewers adjudicated disagreements. The reason for exclusion of studies assessed in the full-text phase is reported in the Appendix A.

### 2.4. Data Extraction and Quality Assessment

Five investigators independently extracted data using a standard data collection form, including study characteristics, country, period of study, setting, technology under investigation, and study population on which it was applied (details are provided in the Appendix A). Where possible, a predefined list of options per variable was created (e.g., study design, study setting, and the aim of the technology). Records included in the review were assessed for their quality based on study design (quantitative, qualitative, mixed methods) with different tools: Cochrane Risk of Bias Tool (RoB2), the Newcastle–Ottawa Scale for case–control studies, the Newcastle–Ottawa Scale for cohort studies, and the Adapted Quality Assessment Tool for before and after studies (details are provided in the Appendix A).

### 2.5. Data Synthesis and Statistical Analysis

The extracted data were reported for qualitative and quantitative analyses. Data recorded using a predefined list of options or short-answer formats were analysed using Microsoft Excel. Data on innovative tools were presented by year, country, setting, and aim of the new technology. Where possible, data were also analysed, comparing traditional and innovative tools for surveillance. For the open-text data column, a thematic analysis was conducted to describe innovations in IPC and to compare the main findings.

## 3. Results

### 3.1. Search Results

The literature search retrieved 1663 articles (Figure 1): 644 from PubMed, 368 from Scopus, and 651 from Web of Science. After the removal of duplicates (n = 1292 articles remaining), the title/abstract screening resulted in the exclusion of 1180 articles. The remaining 112 articles were screened in full text, and 45 were included in the scoping review [10,11,12,13,14,15,16,17,18,19,20,21,22,23,24,25,26,27,28,29,30,31,32,33,34,35,36,37,38,39,40,41,42,43,44,45,46,47,48,49,50,51,52,53,54]. Of these 45 articles, 5 were systematic or scoping reviews. Additional records (n = 325) were identified through checking the references of the included systematic and scoping reviews retrieved from the primary search. Of these, three articles were included in the final version of the scoping review [55,56,57]. In Figure 1, we reported the PRISMA flowchart of the screening phases. The reason for excluding full text articles is given in the Appendix A.

### 3.2. Description of Studies

For the 43 studies included in the analysis [10,11,12,13,14,15,16,17,18,19,20,21,22,23,24,25,26,27,28,29,30,31,32,33,34,35,36,37,38,39,40,41,42,43,44,45,46,47,48,49,55,56,57], most (n = 16) were published in the last year [11,15,17,18,20,21,26,28,29,30,32,39,40,46,48,57], four in 2022 [24,31,34,42], six in 2021 [10,27,33,35,37,49], seven in 2020 [12,14,25,36,38,43,44], six in 2019 [13,22,23,41,55,56], and four in 2018 [16,19,45,47]. Despite the year of publication, we also observed the last year in which the study was conducted. We observed that most of the studies were conducted in 2019 (n = 7, [10,18,20,31,37,41,57]), followed by 2017 (n = 6, [14,22,23,33,44,55,56]), 2018 (n = 6, [17,28,32,34,42,43]), and 2020 (n = 6, [11,15,21,26,27,48]). Only a few studies were conducted in earlier and later years (2021 = 4 [18,24,40,46]; 2022 = 3 [29,30,39]; 2016 = 1 [16]; 2015 = 3 [35,36,38]; 2014 = 1 [19]; 2013 = 2 [12,49]; 2012 = 1 [45]; 2010 = 1 [13]; Unknown = 1 [47]). The mean duration of the studies was 43.7 months (median 38.5; IQR 12–68). Figure 2 reports the detailed number of studies for the year of the end of data collection.

The studies were conducted across multiple continents, with sixteen conducted in North America (fourteen in the US [10,14,16,17,19,20,23,34,38,42,45,46,47,55] and two in Canada [26,35]); twenty-one in Europe (six in the UK, [24,27,29,30,36,57], four in Sweden [12,15,21,49], three in Germany [28,41,44], two in France [13,32] and Italy [37,43], and one each in the Netherlands [56], Spain [40], Norway [39], and Switzerland [11]); five in Asia (three in China [18,25,31], one in Pakistan [48], and one in Thailand [22]), and one in Africa (Rwanda [33]). The majority of studies were single-centre studies (n = 26, [10,11,15,17,18,19,21,22,23,25,28,31,33,34,35,41,42,43,44,45,46,47,48,49,55,57] conducted in tertiary or university hospitals (n = 26, [10,11,13,15,16,17,19,20,21,22,23,25,26,27,30,31,33,34,35,40,42,45,46,48,49,56]). A small number of studies were conducted in general hospitals (n = 12, [12,18,24,28,29,36,37,41,43,44,55,57]), and five studies utilized multiple site record databases at the national or local level [14,32,38,39,47]. No studies were conducted in LTCF. Figure 3 summarizes these results. 

Among the studies, 19 reported data from more or all hospital wards [10,11,12,13,15,16,24,25,29,34,35,36,39,41,44,45,46,49,57], while the rest collected data from specific wards: sixteen from single surgical wards (one paediatric [43], seven orthopaedic [18,23,26,32,40,55,56], three abdominal [21,27,48], one cardiac [30], three neurosurgical [17,22,31], one gynaecology [33]); four from ICUs [19,28,37,42]; and four from other wards [14,20,38,47].

Regarding study design, there were 29 retrospective studies [10,11,12,13,15,17,18,20,21,22,23,25,28,31,32,34,36,37,38,39,40,41,42,43,46,47,48,49,55], two case–control studies [16,35], six prospective studies [19,30,44,45,56,57], four cohort studies [14,24,26,29], one quasi-experimental study [33], and one randomized controlled trial [27].

#### Quality Assessment

Thirty-eight studies (88.37%) were assessed using the Adapted Quality Assessment Tool for retrospective and prospective observational studies (see Quality Appraisal Tools in the Appendix A) [10,11,12,13,15,17,18,19,20,21,22,23,24,25,26,28,30,31,32,34,35,36,37,38,39,40,41,42,43,44,45,46,47,48,49,55,56,57]. Of these, twenty-five were classified as high quality [12,15,17,18,19,20,22,23,24,25,26,28,35,36,37,39,40,43,45,46,47,48,49,56,57], and thirteen as low quality [10,11,13,21,30,31,32,34,38,41,42,44,55]. Three studies (6.98%) were assessed using the Newcastle–Ottawa Scale [14,16,29]. Of these, two were classified as high quality [14,29], and one as fair quality [16]. The remaining two studies (4.65%) were assessed using the Cochrane Risk of Bias (RoB 2) Scale [27,33], and one was judged as a study with a high risk of bias [33], while the other was judged as a study with a low risk of bias [27].

### 3.3. Description of Interventions

#### 3.3.1. Healthcare-Associated Infections

The type of HAIs investigated is reported in 33/43 of the studies included (76.74%). Of these, 24 (72.73%) are surgical site and skin or soft tissue infections [14,15,17,21,22,23,25,26,27,30,31,32,33,35,38,43,46,47,48,55,57], 2 (6.06%) are urinary tract infections [45,49] (± associated with a urinary catheter), 3 (9.09%) are sepsis (± associated with central line catheter) [12,16,28], 1 (3.03%) is ventilator-associated pneumonia [19], and 3 (9.09%) investigate several types of HAIs in the same study [13,20,37].

The microorganisms responsible for the HAIs are reported in 10/43 of the studies included (23.26%). Of these, five investigate more than one microorganism [10,28,36,41,42,44]. The most mentioned microorganisms are VRE [11,28,34,42], MRSA [28,34,36,42], and *P. aeruginosa* [10,34,36,41], followed by *E. coli* [34,36,41], *C. difficile* [34,36,41], and COVID-19 [24,39].

The public health functions explored in our review are surveillance and outbreak detection. The function tackled was outbreak detection in 6/43 studies (13.95%) [10,11,12,24,39,44].

#### 3.3.2. Innovations

Machine learning is the most-applied technology in the included studies (n = 24, 55.81%, [10,11,12,13,14,15,16,18,20,22,24,25,26,31,33,34,35,37,38,40,42,44,46,48]), followed by health informatics technologies (N = 7, 16.28%, [19,28,32,36,39,41,43]) and natural language processing (n = 5, 11.53%, [21,23,45,47,49]). Digital health/e-health/m-health [27,55,56], smartphone and tablet computing devices [29,30,57], and electronic health records (EHRs) [17] are less frequent than the other included studies. Smartphone and tablet computing devices are represented in the UK studies only, while in the US, all digital technology categories, except smartphone and tablet computing devices, are represented.

In the included studies, SSIs are tackled using all digital technology classes. On the contrary, we included two studies focused on urinary tract infections. In both, natural language processing fosters the new technology applied [45,49]. In Figure 4, we show these results.

Proportionally, innovations classified as health informatics are the most representative in terms of individually named/groups of microorganisms (57.14%) [28,36,39,41]. On the contrary, innovations classified as natural language processing, smartphone and tablet computing devices, electronic health records, and digital health/e-health/m-health do not deal with specific microorganisms in the reporting of the studies. These results are reported in the Appendix A.

#### 3.3.3. Thematic Analysis

Several technologies are described in the included studies. From the thematic analysis of the description of the technologies, the potential benefits and the negative impacts on public health functions emerge as three main themes—patient empowerment, workload reduction and cost reduction, and improved sensitivity and personalization:Patient empowerment [29,30,33,56,57]: Smartphone and tablet computing devices with e-health and m-health technologies are implemented, especially in postsurgical settings, to improve patients’ management, fostering their empowerment. These outcomes are also measured in the same studies, with patient-reported experience measures (PREMS) and patient-reported outcomes measures (PROMS).Workload reduction and cost reduction [13,19,21,23,26,28,34,35,39,41,42,43,44,45,46,49]: Health informatics, machine learning, and natural language processing are implemented in various settings. Several articles examine the potential of these technologies in reducing the economic burden of infection and prevention control activities and strengthening the workforce, especially in scarcity situations.Improved sensitivity and personalization [10,11,12,14,15,16,17,18,20,22,24,25,31,37,38,40,42,47,48]: This narrative is recurrent. Machine learning is the digital technology that sustains these expected outcomes in the included articles.

#### 3.3.4. Comparative Analysis

The studies examined different population types (range 69–143,227 patients, mean 17,485.44). This difference is explained through the fact that the purposes of the respective studies were different. In fact, the studies that performed surveillance using machine learning methods (n = 22, [11,12,13,14,15,16,18,20,22,24,25,26,31,33,34,35,37,38,40,42,46,48]) examined larger populations to allow training and testing of the AI algorithm (113–95,858 patients, males 65–47,066, mean 9394.81). The studies that examined smartphone and tablet computing devices and digital health/e-health/m-health (n = 6, [27,29,30,55,56,57]) examined a smaller number of patients (69–1467 patients, males 23–544, mean 364). The latter were used to monitor surgical wounds using images.

We sought a comparative analysis of studies using new technologies versus traditional surveillance methods. We observed that most of the studies did not show this comparison. Only eight studies brought comparative data [11,17,19,27,30,34,39,43]. The studies by Rochon and McLean [27,30] advocated using smartphones to capture remote images for surgical wound surveillance to detect the possible onset of a surgical site infection (SSI). In particular, McLean’s study demonstrated that a digital remote wound follow-up intervention increased the likelihood of diagnosing a surgical site infection fourfold in the early postoperative period. However, it did not reduce the absolute time to the diagnosis of an SSI.

Atkinson and Sundermann employed machine learning for outbreak detection, comparing it with traditional methods [11,34]. Sundermann concluded that the whole-genome sequencing and machine learning algorithm successfully identified outbreaks that traditional surveillance methods failed to detect. On the other hand, the authors emphasized the importance of support for outbreak detection but highlighted that accurate interpretations necessitate data maturity, carefully considering potential confounding factors.

The articles by Atti, Skagseth, Hebert, and Bauer compared surveillance methods for SSIs, COVID-19, and ventilator-associated events (VAEs) [17,19,39,43]. Specifically, Bauer utilized a passive electronic medical record (EMR) algorithm to automatically capture SSIs in spinal fusion cases, comparing its accuracy with administrative medical record review and morbidity and mortality rates. The study argued that the algorithm is more sensitive than traditional methods. Similarly, Hebert employed the algorithm for the automatic detection of VAEs, contending that it simplifies VAE infection prevention by saving time. Atti posited that an algorithm based on the search of regular expressions in unstructured clinical notes is a valuable tool for identifying SSI cases, potentially significantly reducing the workload of conventional surveillance. Skagseth suggested using an automated surveillance system to identify HAI clusters of SARS-CoV-2 in hospitals, comparing them with outbreaks notified through the conventional system. The study concluded that improving preparedness through earlier identification of HAI clusters with automated surveillance can reduce the workload of infection control specialists in hospitals. The complete data extraction form is reported in the Appendix A.

## 4. Discussion

The digital revolution has undeniably permeated various sectors, not least of which is healthcare. The adoption of new technologies is reshaping our approaches to healthcare delivery and prevention [58].

Understanding these innovations, especially in crucial areas such as infectious disease prevention in healthcare settings, is imperative [59]. This scoping review aims to collate and synthesize research literature from 1 January 2018 to 4 November 2023 on the use of digital technologies in the surveillance, control, and prevention of HAIs in hospitals and LTCFs. It also seeks to compare these new technological approaches with traditional surveillance methods.

Echoing a previous scoping review by the European Centre for Disease Prevention and Control (ECDC) [9], we employed a standardized data extraction form to gather study characteristics, country, study period, setting, technologies examined, and the study population. We adhered to ECDC’s definitions for digital technologies relevant to infection surveillance in hospital and LTCF settings, excluding those deemed irrelevant.

Our review identified that most articles provided detailed geographic and temporal contexts of the studies. A noted trend was the time discrepancy between data collection completion and publication year, with a particular increase in articles addressing the subject in 2017. This surge may be linked to advancements in technology, for example, the development of Transformers described by Vaswani et al. [60]. Geographically, while the EU/EEA was a significant focus, the United States led in the overall number of studies. This distribution aligns with the ECDC’s findings and reflects the concentration of research activities in tertiary and academic hospitals.

The primary emphasis in the reviewed literature was on leveraging digital technology for HAI surveillance, especially SSIs. Given the high clinical and economic burden of SSIs [61], alongside their health implications [62], this area has garnered significant attention. Traditional active surveillance methods, which require dedicated infection control staff, are resource-intensive and challenging to sustain universally [63]. Thus, prioritizing SSIs could alleviate financial and workforce pressures. The studies also touched on outbreak detection, validation, and response, albeit to a lesser extent, exploring the impact on clinical decision making and the integration of digital tools into daily practice to understand implications for healthcare professionals [19,57].

Cognitive technologies, as categorized by the ECDC, including machine learning, artificial neural networks, artificial intelligence, natural language processing, and expert systems, were predominant in the literature included. This was followed by device-based technologies, like smartphones and tablets, that facilitate real-time surveillance through applications capable of transmitting images to healthcare professionals, allowing for expedited monitoring of potential infections.

The qualitative analysis has unearthed thematic clusters that shed light on macro-areas of interest for the contribution of new technologies. These include patient empowerment, reduction in the healthcare provider’s workload, cost savings, enhanced sensitivity, and the personalization of strategies. These goals, aimed at the studies that we included, showcase potential applications within their respective contexts. Some of these themes have been previously identified in the ECDC systematic review. Our added discussion highlights how certain challenges, perceived as barriers during the past, could be mitigated [9]. New technologies bring with them the promise of reducing the workload and the burden of costs. There is also a fascinating interplay between the heightened sensitivity of e-health surveillance systems and patient empowerment [30]. This dynamic has the potential to encourage patients to become actively involved in managing their health, with a central focus on enhancing their quality of life [64]. On the other hand, the ever-expanding growth of explainable AI can provide greater assurance regarding the reliability of the outcome provided by the tool. In fact, a deeper understanding of the black box and the results produced can offer healthcare professionals an additional advantage in carefully assessing the available technology and its related utilization [65].

Yet, there lies persistent challenges. Many patients, particularly the most vulnerable, do not possess the necessary resources to engage in this digital shift, thereby raising substantive issues of equity [30]. Additionally, the lack of interdisciplinary leaders capable of introducing novel, discipline-external tools within a healthcare setting remains a bottleneck. These tools are often aimed at specific objectives tied to the implementation of new technology; however, crossing disciplinary boundaries can be a significant hurdle, potentially impeding the swift achievement of shared goals. In this context, it becomes essential to implement training programs for healthcare professionals that incorporate competencies oriented towards understanding and leveraging digital technologies, especially innovative ones. Recognizing this need, numerous international institutions are taking proactive steps, delineating the specific requirements of healthcare professionals within their work environments. Notably, the World Health Organization (WHO) has proposed a European-level action plan, outlining the development of a core competency framework for digital skills within the health workforce [66]. In addition to the necessary expertise, there is a need to use sensitive data for the purpose of these technologies. Machine learning requires large volumes of data to train algorithms. However, creating ad hoc applications or using communication media, such as messaging platforms through which images are transferred, may not guarantee respect for data security and privacy. From this perspective, the necessity arises for a common regulation to address the dual concerns: the development of technologies and the security of data. This requires an adjustment of regulations to account for this new use of data, which, when anonymized, can enhance patient outcomes and research in this field.

### Limitations of this Study

This scoping review has limitations. The search criteria were formulated exclusively in English and confined to the three most-frequently used medical literature search tools. Additionally, we restricted the search to encompass literature published within the previous five years. Nevertheless, given the focus on the novel application of digital technologies for public health purposes, we do not anticipate a significant impact on our findings.

## 5. Conclusions

Our scoping review, encompassing studies from 1 January 2018 to 4 November 2023, provides a comprehensive overview of the advancements in digital technologies for infection control and surveillance within hospital settings. The review indicates a substantial shift towards employing cognitive technologies like machine learning, artificial intelligence, and natural language processing, which augment traditional surveillance methods. Moreover, the development of m-health aims to facilitate real-time surveillance and enables rapid communication between patients and healthcare providers. This approach not only empowers patients in their care but also allows for timely interventions, potentially reducing the incidence and severity of HAIs.

Despite these advancements, our study underscores that the digital divide remains a concern, particularly for vulnerable populations (i.e., elderly people), and highlights the necessity for healthcare organizations to ensure equitable access to digital technologies, considering potential disparities in digital literacy and access among diverse patient populations. Furthermore, the integration of these novel technologies into existing healthcare systems requires interdisciplinary collaboration and leadership, which poses its own set of challenges.

Future research should focus on assessing the long-term impacts of these technologies on patient outcomes, healthcare costs, and overall infection control. The findings of this review suggest that embracing digital innovation in infection control and surveillance could be a key strategy in enhancing healthcare and patient safety.

## Figures and Tables

**Figure 1 antibiotics-13-00077-f001:**
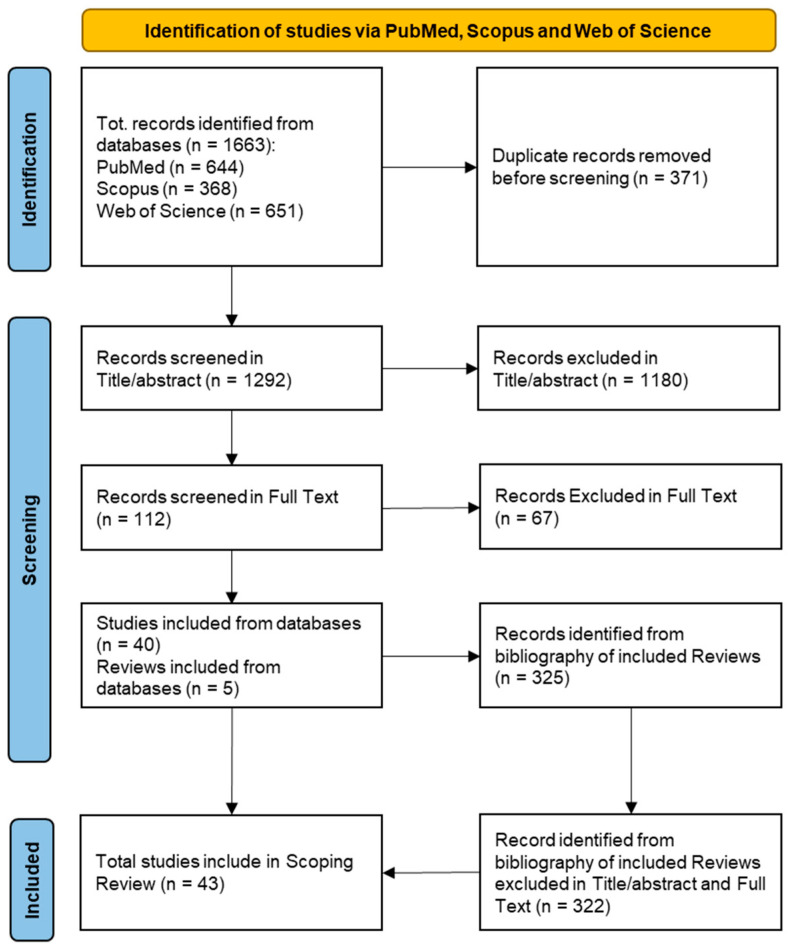
PRISMA flowchart of included articles for scoping review.

**Figure 2 antibiotics-13-00077-f002:**
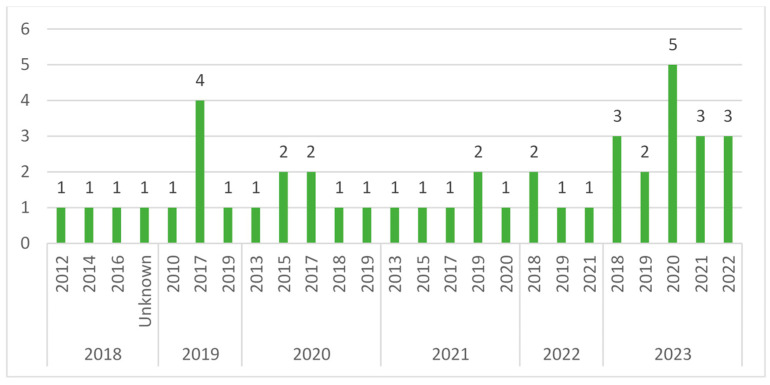
Number of articles for the last year of study grouped by year of publication.

**Figure 3 antibiotics-13-00077-f003:**
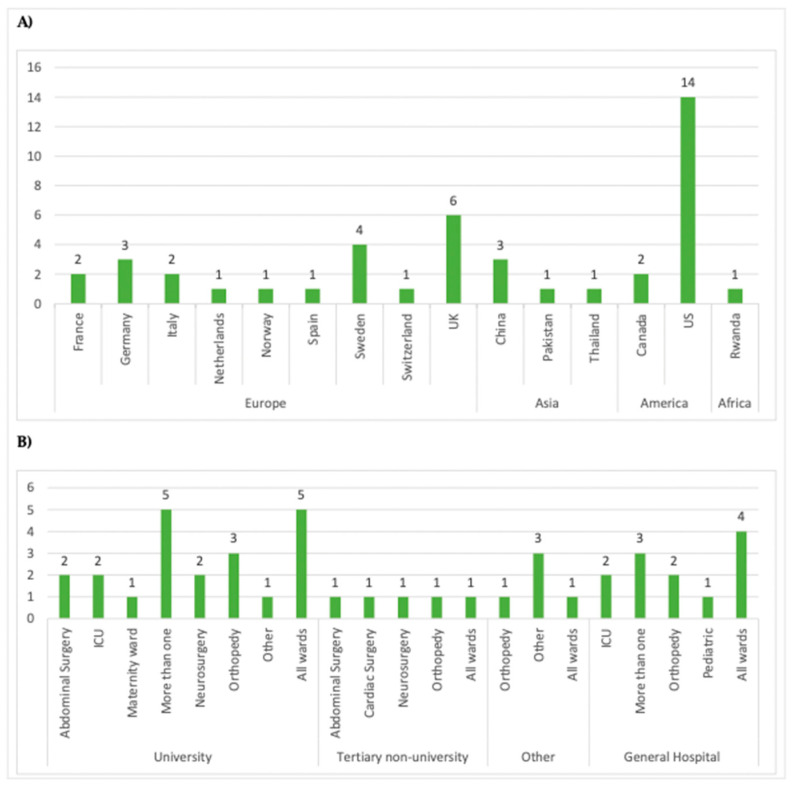
(**A**) Number of articles per country of study grouped by continent. (**B**): Number of articles per ward grouped by setting.

**Figure 4 antibiotics-13-00077-f004:**
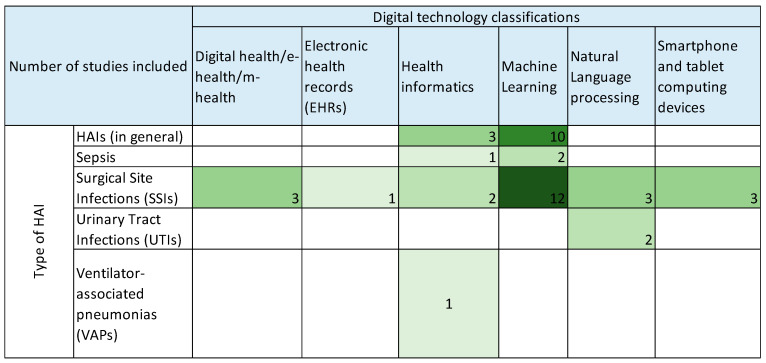
Frequency of HAI type per the six digital technology classes identified in the Methods section.

**Table 1 antibiotics-13-00077-t001:** Inclusion and exclusion criteria for scoping review.

*Variable*	Inclusion	Exclusion
** *Study design/type* **	Randomized controlled trials (RCTs) Nonrandomized, prospective comparative studies Prospective observational studies (e.g., cohort studies) Retrospective observational studies (e.g., case–control studies) Cross-sectional studies Meta-analysis or systematic review Conference proceedings	Narrative review Case reports Non-pertinent publication types (e.g., expert opinions, letters to the editor, editorials, comments, viewpoints) Animal studies Genetic studies, biochemistry, or molecular studies Mathematical modelling studies Study protocols
** *Country* **	All worldwide countries	No exclusion
** *Study subject* **	Innovative technologies for infection control and surveillance	Other types of surveillance
** *Study population* **	Hospitalized individuals (any hospital ward) and LTCF individuals	Non-hospitalized individuals
** *Specific outcomes of interest* **	Qualitative (primary): description of innovations in infection control Quantitative (secondary): evaluation and comparison of accuracy (innovative vs. traditional methods)	Outcomes not related to research question

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
