# Peer review of "Innovative Techniques for Infection Control and Surveillance in Hospital Settings and Long-Term Care Facilities: A Scoping Review"

_antibiotics, 2024, doi:10.3390/antibiotics13010077_

Round 1

Reviewer 1 Report

Comments and Suggestions for Authors

The purpose of these questions is to spark additional thought and investigation. The authors can use the answers to guide them as they expand and improve their article:

1. In the context of AMR, the article extensively covers the role of digital technologies in the surveillance and prevention of Healthcare-Associated Infections (HAIs). However, a crucial aspect not explicitly addressed is how healthcare professionals, particularly those on the front lines, can actively engage with and adapt to these technological changes. Exploring the nuanced impact on clinical decision-making and the integration of digital tools into daily practice would provide a more comprehensive understanding of the article's implications for healthcare professionals. Given the increased emphasis on digital technologies in infection control, how can these innovations address the challenges posed by antimicrobial resistance (AMR) in healthcare settings, and what role do healthcare professionals play in adapting to these changes?

2. While the paper recognizes the prevalence of machine learning in infection control, it does not delve into the potential risks associated with patient data security and privacy. Given the increasing reliance on interconnected health information systems, addressing the challenges and ethical considerations surrounding data sharing, consent, and cybersecurity becomes paramount. A more detailed exploration of these aspects would enhance the comprehensiveness of the article. The article discusses the predominance of cognitive technologies such as machine learning in infection surveillance. Could the authors elaborate on the challenges and ethical considerations related to ensuring the security and privacy of patient data within these advanced technologies, especially in the context of health information exchanges and interoperability?

3. The paper recognizes the prominence of machine learning, but it doesn't explicitly address the critical issue of model interpretability and explainability, which is crucial for gaining trust and acceptance among healthcare professionals. Exploring methodologies and advancements in making these models more interpretable and understandable, perhaps through visualization techniques or transparent algorithms, would provide valuable insights for researchers and practitioners alike. The article highlights the use of machine learning and artificial intelligence in infection surveillance. How can researchers contribute to advancing the interpretability and explainability of these models, ensuring that healthcare professionals can trust and understand the insights derived from these advanced technologies?

4. The paper acknowledges ethical considerations but doesn't explicitly delve into the potential equity challenges associated with the digital divide. Given the growing emphasis on patient empowerment, it's essential to address how healthcare organizations can mitigate disparities in digital literacy and access. Discussing strategies to ensure inclusivity and equitable adoption of these technologies would enhance the ethical discourse within the article. Ethical considerations are briefly mentioned, but how can healthcare organizations ensure equitable access to digital technologies, considering potential disparities in digital literacy and access among diverse patient populations?

5. While the paper hints at potential cost savings, it lacks a detailed discussion on the initial financial investment required for implementing digital technologies. Understanding the economic aspects, including startup costs and ongoing maintenance expenses, is crucial for healthcare managers. Exploring strategies to optimize resource allocation and ensure a sustainable financial model for the integration of these technologies would provide a more holistic view. The article mentions potential cost savings through the use of digital technologies. Can the authors elaborate on the initial investment required for implementing these technologies and the strategies healthcare managers can employ to optimize resource allocation for successful integration?

6. Acknowledging the evolving nature of technology, the article could benefit from a deeper exploration of the regulatory and policy landscape. Policymakers need guidance on navigating frameworks to strike a balance between fostering innovation and safeguarding patient welfare. Examining how policymakers can proactively adapt regulations to keep pace with technological advancements would contribute to a more robust discussion. The article provides insights into the technological landscape but doesn't explicitly discuss the regulatory and policy implications. How can policymakers navigate the evolving regulatory frameworks to ensure the responsible and effective implementation of digital technologies in infection control?

7. While the paper acknowledges the role of education, it doesn't delve into the specifics of developing comprehensive training programs. Health education professionals play a pivotal role in equipping healthcare workers with the necessary skills to navigate and utilize digital technologies effectively. Exploring strategies for curriculum development and continuous training would provide valuable insights for educators and institutions aiming to bridge the digital skills gap in healthcare. The article briefly touches on the potential role of education, but how can health education professionals contribute to developing comprehensive training programs that empower healthcare professionals with the skills needed to effectively utilize these digital technologies?

Author Response

The purpose of these questions is to spark additional thought and investigation. The authors can use the answers to guide them as they expand and improve their article:

  1. In the context of AMR, the article extensively covers the role of digital technologies in the surveillance and prevention of Healthcare-Associated Infections (HAIs). However, a crucial aspect not explicitly addressed is how healthcare professionals, particularly those on the front lines, can actively engage with and adapt to these technological changes. Exploring the nuanced impact on clinical decision-making and the integration of digital tools into daily practice would provide a more comprehensive understanding of the article's implications for healthcare professionals. Given the increased emphasis on digital technologies in infection control, how can these innovations address the challenges posed by antimicrobial resistance (AMR) in healthcare settings, and what role do healthcare professionals play in adapting to these changes?

Response 1: Thank you for this comment. Healthcare professionals play a crucial role in adapting to these technological changes. They are the primary users of these technologies, and their engagement is critical for successful implementation.

The issue of the reduction on the workload and strengthening the workforce, especially in scarcity situations, emerged from the thematic analysis (lines 256-259).

Moreover, other critical issues regarding the role of healthcare professionals are tackled in a residual part of the articles included. Different areas emerged from the included articles:

- Feedback and Usability: Macefield et al. investigated acceptability and usability of a tool for remote assessment of SSIs, not only on the patient’s side, but also on the Healthcare professionals’ side.

- Adaptation and implementation: in Herbert et al., healthcare professionals were involved in the dashboard development and the dashboard evaluation for automated ventilator-associated event electronic surveillance system, and their work was required for a successful process.

We provided to add proper reference in the discussion (lines 331-333)

  1. While the paper recognizes the prevalence of machine learning in infection control, it does not delve into the potential risks associated with patient data security and privacy. Given the increasing reliance on interconnected health information systems, addressing the challenges and ethical considerations surrounding data sharing, consent, and cybersecurity becomes paramount. A more detailed exploration of these aspects would enhance the comprehensiveness of the article. The article discusses the predominance of cognitive technologies such as machine learning in infection surveillance. Could the authors elaborate on the challenges and ethical considerations related to ensuring the security and privacy of patient data within these advanced technologies, especially in the context of health information exchanges and interoperability?

Response 2: Thank you for this comment. We add a paragraph to explain this topic (lines 369-377).

  1. The paper recognizes the prominence of machine learning, but it doesn't explicitly address the critical issue of model interpretability and explainability, which is crucial for gaining trust and acceptance among healthcare professionals. Exploring methodologies and advancements in making these models more interpretable and understandable, perhaps through visualization techniques or transparent algorithms, would provide valuable insights for researchers and practitioners alike. The article highlights the use of machine learning and artificial intelligence in infection surveillance. How can researchers contribute to advancing the interpretability and explainability of these models, ensuring that healthcare professionals can trust and understand the insights derived from these advanced technologies?

Response 3: Thank you for your feedback. We have incorporated a brief comment (lines 351-355) in the text, clarifying that these models require ongoing research in the realm of interpretability and explainability. This is essential for achieving a more comprehensive understanding of the black box and the outcomes it generates. However, we would like to emphasize that our review does not specifically delve into detailing various techniques for enhancing explainability and interpretability.

  1. The paper acknowledges ethical considerations but doesn't explicitly delve into the potential equity challenges associated with the digital divide. Given the growing emphasis on patient empowerment, it's essential to address how healthcare organizations can mitigate disparities in digital literacy and access. Discussing strategies to ensure inclusivity and equitable adoption of these technologies would enhance the ethical discourse within the article. Ethical considerations are briefly mentioned, but how can healthcare organizations ensure equitable access to digital technologies, considering potential disparities in digital literacy and access among diverse patient populations?

Response 4: Thank you for your enlightening comment. Healthcare organizations play a pivotal role in ensuring equitable access to digital technologies. By adopting a multi-faceted approach that addresses digital literacy, access disparities, and cultural and linguistic inclusivity, these organizations can help mitigate the equity challenges associated with the digital divide. However, only few authors included this pivotal aspect in their research. Rochon et al. acknowledged digital divide as a barrier for the implementation of their intervention.

We added proper reference at line 331 in the discussion and we enriched the conclusion with the role of healthcare organizations in addressing inequities.

  1. While the paper hints at potential cost savings, it lacks a detailed discussion on the initial financial investment required for implementing digital technologies. Understanding the economic aspects, including startup costs and ongoing maintenance expenses, is crucial for healthcare managers. Exploring strategies to optimize resource allocation and ensure a sustainable financial model for the integration of these technologies would provide a more holistic view. The article mentions potential cost savings through the use of digital technologies. Can the authors elaborate on the initial investment required for implementing these technologies and the strategies healthcare managers can employ to optimize resource allocation for successful integration?

Response 5: Thank you for raising this important point. The need for a more comprehensive analysis of the economic aspects of implementing digital technologies in healthcare it is fundamental. However, the included articles do not delve into initial investment analysis, cost-effectiveness analysis, and strategies for optimizing resource allocation.

  1. Acknowledging the evolving nature of technology, the article could benefit from a deeper exploration of the regulatory and policy landscape. Policymakers need guidance on navigating frameworks to strike a balance between fostering innovation and safeguarding patient welfare. Examining how policymakers can proactively adapt regulations to keep pace with technological advancements would contribute to a more robust discussion. The article provides insights into the technological landscape but doesn't explicitly discuss the regulatory and policy implications. How can policymakers navigate the evolving regulatory frameworks to ensure the responsible and effective implementation of digital technologies in infection control?

Response 6: Thank you for raising this crucial point. We added a paragraph (lines 369-377) to explain the importance of a regulatory framework to balance innovation, patient safety and privacy. We decided to not include an overview of key regulations, agencies involved, and the types of approvals required for new technologies, because our review do not delve into this specific matter. However, we totally agree with you about the importance of exploring the regulatory and the policy landscape.

  1. While the paper acknowledges the role of education, it doesn't delve into the specifics of developing comprehensive training programs. Health education professionals play a pivotal role in equipping healthcare workers with the necessary skills to navigate and utilize digital technologies effectively. Exploring strategies for curriculum development and continuous training would provide valuable insights for educators and institutions aiming to bridge the digital skills gap in healthcare. The article briefly touches on the potential role of education, but how can health education professionals contribute to developing comprehensive training programs that empower healthcare professionals with the skills needed to effectively utilize these digital technologies?

Response 7: Thank you for this comment. Let's add a paragraph in the discussion to expand on this issue.

Reviewer 2 Report

Comments and Suggestions for Authors

The study is interesting, but the authors should be more specific in showing the significance of new technologies in the preventable surveillance of healthcare associated Infection. 

Author Response

The study is interesting, but the authors should be more specific in showing the significance of new technologies in the preventable surveillance of healthcare associated Infection. 

Response: Thank you for your comment. To be more specific, the Quality Assessment of the studies included was performed. We added a paragraph in the last part of the Result section (lines 199-208). Moreover, we expanded the manuscripts focusing on the role of healthcare workforce (lines 331-333), considerations on privacy matter (lines 369-377), the role of healthcare organizations in addressing inequities (lines 394-398).

Reviewer 3 Report

Comments and Suggestions for Authors   In the present scoping Review, the authors present the implementation of new technologies into healthcare settings, by following the PRISMA-ScR guideline, searching for studies of new digital technologies applied to the surveillance, control, and prevention of HAI in hospitals and LTCFs published from 2018  to November 4th, 2023. Comparative analysis between new technologies and traditional methods showed different population types, with machine learning methods examining larger populations for AI algorithm training. They conclude that while digital tools show promise in HAI surveillance, there are also limitations, including challenges in resource distribution and interdisciplinary integration in healthcare settings.  The methodology is adequate and appropriate references are included. The presentation of the results needs improvement. The Discussion is consistent with the results. Major comments: - lines 162-182: The data may be presented more clearly by using pie chats for the percentages/numbers of studies by continent/ country, hospitals/wards -Figure 2: Please use a grouped bar graph for the year of publication  and the last year the study conducted. --Please use italics for the species names.   

Author Response

In the present scoping Review, the authors present the implementation of new technologies into healthcare settings, by following the PRISMA-ScR guideline, searching for studies of new digital technologies applied to the surveillance, control, and prevention of HAI in hospitals and LTCFs published from 2018 to November 4th, 2023. Comparative analysis between new technologies and traditional methods showed different population types, with machine learning methods examining larger populations for AI algorithm training. They conclude that while digital tools show promise in HAI surveillance, there are also limitations, including challenges in resource distribution and interdisciplinary integration in healthcare settings.  The methodology is adequate and appropriate references are included. The presentation of the results needs improvement. The Discussion is consistent with the results. Major comments: - lines 162-182: The data may be presented more clearly by using pie charts for the percentages/numbers of studies by continent/ country, hospitals/wards -Figure 2: Please use a grouped bar graph for the year of publication and the last year the study conducted. --Please use italics for the species names.   

Response: Thank you for this comment. As you suggested, we add a grouped bar graph for the year of publication and the last year the study conducted in Figure 2 (line 166). We used a bar graph also for the new Figure 3 that represents the distribution of the studies for countries and for hospital setting (line 184). We also changed in italics the species names.

Round 2

Reviewer 1 Report

Comments and Suggestions for Authors

I appreciate your answers to the queries posed. Even though your explanations cover the majority of the issues, the article could benefit from more detail on specific approaches for healthcare professionals to engage, specific risk mitigation for data security, the interpretability and transparency of machine learning models, and real-world examples of addressing the digital divide and economic issues. Further investigation into health education initiatives and regulatory systems may yield more thorough insights. Even with these possible areas for improvement, I think the work is ready to be published and will substantially contribute to the field. We appreciate your careful consideration of our suggestions to improve the quality of your article and your intelligent comments.

Reviewer 2 Report

Comments and Suggestions for Authors

The authors have responded to the comments and improved the manuscript